# New Role of the Serotonin as a Biomarker of Gut–Brain Interaction

**DOI:** 10.3390/life14101280

**Published:** 2024-10-09

**Authors:** Hong Nian Liu, Masanao Nakamura, Hiroki Kawashima

**Affiliations:** Department of Gastroenterology and Hepatology, Nagoya University Graduate School of Medicine, 65 Tsurumai-cho, Showa-ku, Nagoya 466-8550, Japan; makamura@med.nagoya-u.ac.jp (M.N.);

**Keywords:** serotonin, gut–brain interaction, biomarker, mood, irritable bowel syndrome (IBS), functional dyspepsia (FD), inflammatory bowel disease (IBD), cancer, gut homeostasis

## Abstract

Serotonin (5-hydroxytryptamine: 5-HT), a neurotransmitter that regulates mood in the brain and signaling in the gut, has receptors throughout the body that serve various functions, especially in the gut and brain. Selective serotonin reuptake inhibitors (SSRIs) are used to treat depression, but their efficacy is uncertain. Depression is often associated with early gastrointestinal symptoms. Gut disorders such as functional dyspepsia (FD), irritable bowel syndrome (IBS) and inflammatory bowel disease (IBD), including ulcerative colitis (UC) and Crohn’s disease (CD), are linked to elevated serotonin levels. In this review, we would like to discuss the approach of using serotonin as a biomarker for gut–brain, and body-wide organ communication may lead to the development of preventive and innovative treatments for gut–brain disorders, offering improved visibility and therapeutic monitoring. It could also be used to gauge stress intensity for self-care and mental health improvement.

## 1. Introduction

Serotonin (5-hydroxytryptamine: 5-HT), also called the “happy chemical”, is present in 95% of the gastrointestinal tract, and selective inhibitors for serotonin re-uptake transporters are known to treat depression [1,2,3,4,5]. 5-HT, a monoamine neurotransmitter involved in mood control in the brain, also plays a critical role in cellular signaling in the gut. For example, 5-HTR4 on enteric neurons is involved in neurogenesis through a pathway in which intracellular cyclic adenosine monophosphate production activates proteins involved in cellular proliferation, namely protein kinase A and the extracellular signal-regulated kinase pathway [6]. 5-HT is an important gastrointestinal signaling molecule. The great amount of 5-HT in the gastrointestinal (GI) system plays a central role in the regulation of processes such as GI secretion, peristalsis, and vasoconstriction, among others [7]. In addition, some enteric neurons in the descending peristaltic reflex pathway also release 5-HT as a neurotransmitter. Enteric neurons and smooth muscle cells express various 5-HT receptors depending on cell type and location and their functions are affected by this signaling molecule [3,6,7,8,9,10,11,12]. Therefore, 5-HT receptors are important targets for pharmacological interventions in gut functional and psychiatric disorders.

M. M. Rapport is credited with isolating serotonin from blood and describing its structure as 5-HT in 1948 [13]. 5-HT exerts diverse operations in the GI tract through distinct receptors. Serotonin receptors can be divided into seven types (15 subtypes) from 5-HT1 to 5-HT7, and the intestine expresses five of the seven identified 5-HT receptor families: 5-HT1, 5-HT2, 5-HT3, 5-HT4, 5-HT5, 5-HT6 and 5-HT7. It influences a variety of bodily functions [3,9,11]. There are two diseases in the gut: one is irritable bowel syndrome (IBS) and the other is inflammatory bowel disease (IBD).

IBS is a condition that affects a significant portion of the adult population worldwide, with estimates ranging from 10 to 23% [14,15]. Also, in Japan, according to the Ministry of Health and Welfare, it is estimated that by 2020, 10–15% (12–18 million) of the population, or 1 in 10 people, will experience symptoms of IBS. IBS is a significant healthcare burden across settings and geographies. Accurate case definition in IBS remains difficult due to the high frequency of symptoms in the community, variations in diagnostic criteria and the stringency with which they are implemented, the lack of specific histopathologic changes, and the lack of a definitive point of onset [16,17]. The crucial role of 5-HT as a neurotransmitter and paracrine signaling molecule in the GI tract is well established. The gut and the brain are in a constant two-way interaction in which the 5-HT pathway is involved. Recent studies have implicated alterations in 5-HT signaling in dysfunction. It has been suggested that IBS may be linked to serotonin, a condition that is often associated with depression and anxiety [17,18]. Serotonin has also been reported to be associated with IBD [3,19,20]. IBD includes ulcerative colitis (UC) and Crohn’s disease (CD). Both are chronic inflammatory bowel diseases. UC affects the lining of the large and small intestines, causing symptoms like diarrhea, abdominal pain, and bloody stools. CD affects the entire GI tract, from the mouth to the anus [21,22]. IBD is common in Western countries, especially in North America and Europe, and UC and CD have been recognized as occidentalized diseases due to their higher incidence and prevalence rates in occidental countries. CD usually presents between 20 and 30 years of age, whereas UC usually presents between 30 and 40 years of age, as well as from 60 to 70 years of age [3]. In Japan, the Ministry of Health and Welfare designated IBD as a nationally recognized refractory UC, which affects 130,000 patients, and CD, which affects approximately 50,000 people, and early detection, diagnosis and treatment are important to control disease progression. Serotonin transporter (SERT) is a type of monoamine transporter protein that transports the neurotransmitter serotonin from the synaptic cleft back to the presynaptic neuron. mRNA expression is reduced in patients with active UC and CD [23]. During these inflammatory processes, 5-HT signaling and SERT function become abnormal [24,25]. Evidence suggests that the prevalence of depression ranges from 15% to 25%, with rates possibly lower in UC patients than in those with CD. Anxiety is even more prevalent, with rates approaching 30% in IBD patients. Rates of anxiety and depression were higher during periods of flare-up [3].

Currently, the interstitial cells of Cajal (ICCs: act as pacemaker cells in the GI tract) are the object of study in many medical fields to understand intestinal motility and its involvement in the development of GIST pathology, as well as to elucidate the pathogenesis of various motility disorders [26,27]. Cajal cells are named after the great Spanish neuroanatomist Santiago Ramón y Cajal, winner of the Nobel Prize in Physiology/Medicine in 1906.

Oscillations of [Ca^2+^]i (“i” refers to ions within the intracellular space) in ICCs have been considered as the primary mechanism of GI peacemaking, in which Ca^2+^-dependent ion channels are periodically activated to generate pacemaker potentials [28,29]. We have previously provided evidence that 5-HT regulates the pacemaker activity of ICC and found that 5-HT enhances both Ca^2+^ and electric activities of ICCs via the 5-HT3 receptor, a nonselective cation channel permeable to Ca^2+^, and so ICCs appears to be a promising target for functional motility disorders in the gut [30]. Therefore, 5-HT3 receptors in the gut may contribute to gut–brain interactions, and our findings suggest that the 5-HT modulation of ICC activity should also be considered in gut motility disorders, for example, IBS [31,32].

Although these results have been confirmed in mouse ileal ICCs, we hypothesize that 5-HT may be related to human gut diseases such as IBS and IBD, but how it is related still needs to be proven in clinical experiments.

To test this hypothesis, we recently measured serum levels of 5-HT in patients with Crohn’s disease using a 5-HT ELISA kit that measures 5-HT levels in 1:16 dilutions of human serum (samples were obtained from 52 patients with Crohn’s disease, 39 of whom were in the SES-CD (simple endoscopic score for Crohn’s disease) active disease group and 13 of whom were in remission). The results demonstrated that 5-HT levels were significantly elevated in patients with active SEC-CD compared to those in remission (data omitted).

Some studies have reported elevated blood and urinary serotonin levels and serotonin secretion in patients with unclassified irritable IBS-U, UC and CD [33,34,35,36,37].

In a CD study (serum samples were obtained from a German cohort of 96 CD and UC patients), the authors first measured tryptophan (TRP), 5-HT and kynurenine (KYN) in the serum of patients with UC and CD and compared patients in different disease categories. In patients with UC, none of the metabolites differentiated between patients in remission, active patients, or refractory patients. In CD, there were no differences in TRP or KYN levels between disease categories, but 5-HT levels were significantly elevated in active patients [33].

Circulating CRP (C-reactive protein) is one of the most widely used biomarkers of increased disease activity in IBD. The authors next compared CRP levels in UC and CD patients at different stages of the disease. In UC (remission N = 14, active N = 13, refractory N = 13) patients, CRP levels were lower in patients in remission than in patients with active or refractory disease. However, there was no difference in CRP levels in CD (remission N = 15, active N = 14, refractory N = 16) patients. These data suggest that new noninvasive biomarkers are needed to better understand disease activity in UC and CD.

Further, the authors compared CRP and 5-HT to see if they could tell if someone had active UC or CD. There was no difference in CRP levels between the two. In contrast, serum 5-HT levels were markedly increased in active CD when compared with 5-HT levels in active UC. These results show that 5-HT is more elevated in active CD than in CRP. In addition,, SERT immunoreactivity was decreased in the ileum and colon of patients with CD compared to healthy controls. These data agree with previous findings that SERT expression is lower during inflammation. This may explain the higher levels of circulating 5-HT during active CD [33].

In the UC study, 5-HT levels were measured by ELISA in patients with UC (75, 20–64 years old, mean 38 ± 11 years old) and in a control group (100, 20–64 years old, mean 38 ± 11 years old). The results showed that UC patients had significantly elevated 5-HT levels than healthy controls (*p* < 0.01) [37].

Recent IBS studies have looked for non-invasive biomarkers in serum, breath and fecal material. Biomarkers should be based on biological and pathogenic processes, but most IBS biomarkers have been developed to identify other diseases and are therefore not applicable to IBS. Biomarkers have been developed to improve diagnosis, differentiate IBS from other diseases, and identify subtypes of IBS. True mechanistic biomarkers will help diagnose IBS, not just rule out other diseases.

Although new serologic biomarkers have been developed for the treatment of diarrheal IBS using rat models and results from large clinical trials, their efficacy in treating diarrheal IBS has been limited. However, their effectiveness in diagnosing IBS and prognosis and whether they work well or not is unknown [38].

Also, unlike inflammatory markers such as CRP, ESR (erythrocyte sedimentation rate) and circulating cytokines, more than 95% of systemic 5-HT is produced by the gastrointestinal tract, which may be beneficial in differentiating IBD states [33]. These results are sufficient to demonstrate the feasibility, practicality and strategic importance of using serotonin as a biomarker for future studies. Here, we would like to highlight the potential utility of serotonin as a biomarker of gut–brain interactions in IBS and IBD.

This review addresses how the use of serotonin as a noninvasive biomarker of gut–brain and systemic organ communication may lead to the development of innovative treatments for gut–brain disease prevention, improved visualization, and therapeutic monitoring. It could also be used to gauge stress intensity for self-care and mental health improvement.

## 2. Methods

We searched the Web of Science, PubMed, Google Scholar Science, the World Health Organization (WHO), the Ministry of Health, Labor and Welfare of Japan, Japanese journals, and a variety of resources to describe the effects of serotonin biomarkers on the prevention, diagnosis, efficacy assessment, monitoring of disease activity, and prognostic assessment of gut–brain disorders. In addition, we analyzed the literature to gain insight into the current state of research on gut–brain interactions. This will help scientists interested in understanding the key messages and research frontiers in the field and the feasibility of using serotonin as a biomarker for gut–brain interactions, and we present the latest trends in biomarkers for gut–brain interactions (Table 1).

## 3. Depression

The exact mechanism of depression in the brain is still not fully understood, but low levels of 5-HT are associated with depression; depression is a mood disorder in which people experience persistent feelings of low mood, sadness, or anger [1,2,3,4,5].

Depression is the most common mental disorder, and its prevalence is increasing. It is estimated to affect approximately 10% of the population. Depression is a common and serious illness characterized by symptoms such as a lack of pleasure, low energy, sadness, loss of appetite and difficulty sleeping. Although the average age of onset of depression is usually between 30 and 40 years, the disease occurs in children, adolescents, and older adults. Data show that pre-adolescent boys are more likely to suffer from depression than women. This trend reverses during adolescence, with females about twice as likely as males to experience a depressive episode in the following years. This is a very complex disorder that is difficult to fully understand and is one of the greatest challenges facing the fields of medicine and neuroscience [39,40,41]. Clinical and preclinical data have allowed scientists in the field of depression research to formulate a number of hypotheses to explain the causes and development of the disorder. It is believed that depression is the result of reduced serotonin levels in brain structures responsible for emotional and cognitive processes [42,43,44,45]. Symptoms of depression vary from person to person, but if they persist for more than two weeks, it is important to seek a professional diagnosis, have your blood serotonin levels tested, and consult a psychiatrist or psychologist. It is important not to let the condition become worse.

## 4. Serotonin Syndrome Serotonin Toxicity

However, a high level of 5-HT is associated with anxiety, aggression, and reduced tolerance to stress, as well as serotonin syndrome. In Japan, according to the Ministry of Health and Welfare, it is estimated that by 2021 (summary), serotonin syndrome/serotonin toxicity will be a possible side effect of antidepressants and other serotonin-related medications. Symptoms include altered mental status, such as anxiety, restlessness, anxiety, restlessness, agitation, and delirium, and autonomic hyperactivity, such as tachycardia, high blood pressure, high fever, sweating, tremors, vomiting, diarrhea. It is usually caused by an increased stimulation of serotonin receptors in the brain by medication. Specific examples include the use of medications for therapeutic purposes, an overdose of certain medications, and unintentional drug interactions when two medications that stimulate serotonin receptors are taken at the same time. Symptoms of serotonin syndrome: in most cases, serotonin syndrome occurs within 24 h of changing or starting medication, usually within 6 h. Early detection and early response points: after starting or increasing the dose of a drug with serotonin-related effects, a sudden onset of adverse effects should be suspected if symptoms such as mental agitation, convulsions, and rapid pulse occur. Antidepressants are the most common drugs that cause serotonin syndrome.

In particular, common selective serotonin reuptake inhibitors (fluvoxamine, paroxetine, sertraline, and escitalopram), often referred to as selective serotonin reuptake inhibitors (SSRIs), are the drugs that most commonly cause serotonin syndrome. Mood stabilizers, anti-anxiety medications, narcotic pain relievers, and migraine medications may also contribute to the condition. This is especially true for people who take several antidepressants or other medications at the same time [6,39].

## 5. 5-HT Receptors and Primary Functions

(1) The 5-HT1 receptor and its subtypes (5-HT1A, 5-HT1B, 5-HT1D, 5-HT1E, 5-HT1F, and 5-HT1P) are in enteric neurons, smooth muscle, interstitial cells of Cajal and enterocytes and exert a serotonin bioavailability-dependent effect on the relaxation of the gastric fundus in response to a prokinetic agent. The 5-HT1A receptor provides a pharmacological target for anxiolytics, and the 5-HT1B/1D 5-HT1F receptor is an antimigraine target.

(2) The 5-HT2 receptor and its subtypes (5-HT2A, 5-HT2B, and 5-HT2C) are in enteric neurons, smooth muscle, and interstitial cells of Cajal. Many antipsychotics interact with 5-HT2 receptors at clinically relevant concentrations, and an antagonism of 5-HT2A receptors may contribute to the therapeutic effect.

(3) The 5-HT3 receptor and its subtype (5-HT3 and 5-HT3A) are ligand-controlled ionic channels, located in enteric neurons, interstitial cells of Cajal, enterocytes, extrinsic nerves, and EC cells. Antagonists such as ondansetron can provide effective antiemetic therapy as vomiting may occur following aggressive anticancer therapy. 5-HT3 receptors also mediate GI prokinetic activity in patients with IBS or carcinoid tumor in both types of patients, and 5-HT3 receptor antagonists are effective in reducing diarrhea.

(4) The 5-HT4 receptors are expressed in CNS neurons, enteric neurons, and EC cells. In the GI tract, they accelerate intestinal motility by increasing the release of presynaptic membrane acetylcholine. In EC cells, they induce serotonin release. This receptor increases gastrointestinal motility by enhancing the activity of, among others, cholinergic neurons in the gastrointestinal tract and is activated by certain anti-constipation drugs.

(5) The 5-HT5 receptors are GPCRs, the family has two members, 5-HT5A and 5-HT5B receptors. The latter receptor is non-functional in humans due to the insertion of a stop codon in the gene, resulting in a truncated protein. Little is known about the 5-HT5A receptor; it is not actively targeted for therapeutic benefit.

(6) Both 5-HT6 and 5-HT7 receptors are G protein-coupled receptors (GPCRs) that bind via Gs and promote cAMP synthesis via adenylate cyclase [3,6,7,8,9,10]. These two receptors have been of great interest as potential clinical targets. The structure and function of 5-HT will continue to be further elucidated (Figure 1, Table 2).

## 6. Co-Operation between Serotonin and Dopamine Receptors Is Key to Prevention of Gut–Brain Disorders

In addition, serotonin plays a critical role in the proper functioning of dopamine. Dopamine is one of the most important signaling molecules regulating the central nervous system and the enteric nervous system. Of the five different dopamine receptors (D1, D2, D3, D4, and D5), the D1, D2, and D5 receptors are expressed in the GI tract, with the highest mRNA expression of the D2 receptor, which is involved in the regulation of GI motility [48,49]. The loss of dopamine leads to Parkinson’s disease (PD), and patients with PD typically experience GI symptoms such as dysphagia, nausea, constipation and peristaltic dysfunction [50]. Because dopamine plays an important role in GI function, including exocrine and peristaltic movements, dopaminergic deficiency is thought to be responsible for GI symptoms in PD patients [51,52,53]. Some studies have indicated that people with PD may begin to experience gut symptoms, such as constipation, sleep disturbances, and depression, as early as 20 years prior [54,55]. Studies have also shown that IBD can increase the incidence of PD by 22% to 25%. By improving the symptoms of IBD and gut-related issues, it may be possible to prevent the onset of Parkinson’s disease. This is due to the strong connection between the gut and the brain, also known as the gut–brain interaction. In particular, an overdose of dopamine can in turn lead to schizophrenia. Interestingly, like the D2 receptor involved in gut motility regulation, antipsychotic drugs for schizophrenia act primarily by occupying the dopamine D2 receptor and have been shown to be effective in controlling positive symptoms. Some studies have shown that 5-HT and dopamine cooperatively control the peristaltic movement of the gut [56].

In 2018, our study showed that dopamine receptor D2, together with serotonin receptors 5-HT2 and 5-HT3, is involved in the maintenance of [Ca^2+^]i oscillations in the ICCs [57]. Serotonin and dopamine are two important neurotransmitters that affect mental health. They are also important for gut health. Thus, like serotonin, dopamine can be used as a biomarker in the gut–brain as well as in related organs throughout the body. This approach allows for the early prediction and prevention of PD and schizophrenia, and human serotonin and dopamine levels could potentially be quantified and visualized in the gut–brain and related organs throughout the body (Figure 2).

## 7. Limitations of Endoscopy and the Inevitability of 5-HT as a Gut–Brain Biomarker

IBD is a long-term digestive disease. Endoscopy continues to offer significant potential for improving the long-term prognosis of patients with IBD. However, endoscopy is an invasive test, and the frequent use, complexity, and cost of endoscopy can present a psychological, physical, and financial burden for the patient, as well as an inconvenience for the physician. Medications can control symptoms and prevent flare-ups, but they can also cause side effects. These may include abdominal pain, fever, joint pain, and blood in the stool [5,33]. Steroids are effective during flare-ups and are recommended for short-term use. However, long-term use can produce side effects, weakening anti-inflammatory properties and leading to intestinal inflammation and depression in 30% of IBD patients [4,5]. New, cheaper, and more accurate methods for the early diagnosis and prognostic monitoring of IBD are needed. A minimally invasive blood test using serotonin as a noninvasive biomarker may help in the early diagnosis of the disease, assessment of treatment efficacy, monitoring of disease activity and prognostic assessment.

Fortunately, recent studies have shown that 5-HT can be used as a biomarker for the early diagnosis and prognosis of the disease [33,58]. The pandemic of coronavirus disease (COVID-19) is associated with high mortality rates worldwide (WHO Report, 1 May 2024). COVID-19 is an infectious disease caused by the SARS-CoV-2 virus. Most people infected with the virus will experience mild to moderate respiratory illness and recover without requiring special treatment. However, some will become seriously ill and require medical attention. Older people and those with underlying medical conditions such as cardiovascular disease, diabetes, chronic respiratory disease, or cancer are more likely to develop a serious illness. Anyone at any age can become sick with COVID-19 and become seriously ill or die at any age. The best way to prevent and slow down transmission is to be well informed about the disease and how the virus is spread.

Due to its complex pathophysiology, diagnostic and prognostic biomarkers for effective patient management remain scarce [58]. 5-HT levels in the serum of COVID-19 patients were tested. Serum 5-HT levels were found to be reduced in patients with more severe disease and also significantly lower in patients who had moderate disease but subsequently worsened compared to patients who did not have severe disease. Serum serotonin levels may be a valuable biomarker of COVID-19 severity and prognosis.

This demonstrates the feasibility of serotonin as a biomarker for early disease detection and prognosis.

## 8. Symptoms of Functional Dyspepsia (FD) and 5-HT Receptor

We already know that there are diseases in the gut, IBS and IBD, and there is another disease called functional dyspepsia (FD), which is also related to serotonin. FD is a condition in which upper abdominal symptoms appear to originate in the stomach or duodenum, but no organic disease such as ulcers or cancer is found during tests such as upper gastrointestinal endoscopy. However, symptoms of pain and discomfort persist.

Although functional dyspepsia is not an organic disorder that can cause these symptoms, four symptoms are thought to originate in the stomach and duodenum. Symptoms that originate in the stomach and duodenum include one or more of the following: upper abdominal pain, upper abdominal burning, abdominal heaviness after meals and early satiety (the feeling of fullness in the stomach immediately after starting a meal), and an inability to finish the meal. In addition, symptoms were defined as having been present for more than 6 months, occurring several times per week, and persisting for 3 months. Functional dyspepsia affects approximately 21% of the population worldwide. The cause of functional dyspepsia is unknown, but psychological stress, abnormal gastrointestinal motility, allergies, and serotonin are thought to contribute [59,60,61].

### 8.1. Symptoms of Functional Dyspepsia and Serotonin Transporters (SERTs)

The research team used PET to quantitatively analyze the serotonin transporter binding capacity of different brain regions in FD patients. They used questionnaires to survey nine FD patients (the FD group) and eight healthy subjects (the control group) about FD, depression, and anxiety symptoms. Considering this in conjunction with the PET analysis results, they found that the FD group had moderate significance: increased serotonin transporter binding capacity was observed in the brain and thalamus [59].

When they next analyzed the correlation between the same connectivity capacity of the midbrain, thalamus and hippocampus and individual symptoms, the team found that in the midbrain, Gastrointestinal Symptom Rating Scale (GSRS) scores were positively correlated with abdominal pain; in the thalamus, there was a positive correlation between GSRS scores and abdominal pain and backward tilt; and in the hippocampus, there was a correlation between abdominal pain and anxiety symptoms. They concluded that the upregulation of SERT levels in the midbrain and thalamus may underlie the pathogenesis of FD, such as abdominal and psychological symptoms, via gut–brain interactions [59].

The up-regulation of SERT levels in the midbrain and thalamus may underlie the pathogenesis of FD, such as abdominal and psychological symptoms, through the gut–brain-interaction [59].

### 8.2. Symptoms of Functional Dyspepsia and 5-HT Receptor

Recent studies have shown that 5-HT4 receptor agonists stimulate late contractions of gastric smooth muscle [60]. 5-HT4 receptors, which are primarily located in the gut, have been shown to increase pre-and postprandial gastric volume in healthy individuals and to impair gastric peristalsis and sensory function in patients with FD. 5-HT4 receptor agonists have also been shown to significantly attenuate colon-induced visceral pain in FD patients, regardless of their *Helicobacter pylori* (*H. pylori*) or gastritis status. A 5-HT1 agonist was shown to improve abdominal symptom scores in patients with FD, whereas a 5-HT1A agonist, R-137696, failed to improve symptoms or visceral hypersensitivity in patients with FD. A meta-analysis of 10 studies showed that patients with FD treated with 5-HT1 and 5-HT4 agonists had significantly higher response rates and symptom improvement compared with the placebo group [61,62].

## 9. 5-HT Involved in Carcinogenesis

Several studies have demonstrated serotonin’s growth-stimulatory effect on several types of cancers, carcinoids and other tumor cells. Here, we focus on typical digestive tract cancers whose prevalence is still increasing, such as the following: colorectal cancer, carcinoid tumors (CSs), cholangiocarcinoma, and hepatocellular carcinoma (HCC) [63,64].

### 9.1. Colorectal Cancer

Colorectal cancer is one of the most common causes of cancer deaths worldwide. Since it was discovered that serotonin controls the growth of colorectal cancer cells, scientists have learned a lot about how serotonin affects the growth of intestinal stem cells and the development of colon tumors [64,65]. Recent studies have revisited the involvement of the serotonin receptor in colorectal cancer and suggest a potential role of 5-HT1, 5-HT3, and 5-HT4 receptors in this tumor, particularly the 5-HT1D, 5-HTR3C, and 5-HTR4 receptor subtypes, and in experimental mouse models, serotonin levels are higher in colorectal cancer patients and correlate with poorer cancer prognosis [64,65].

Another study showed that colon cells from TPH1 knockout animals have more DNA damage and more severe intestinal inflammation than those from wild-type animals. Both are etiologic precursors of colon cancer [66]. This evidence implicates 5-HTR activation in cancer. Studies show that high levels of serotonin in mice activate lymphocytes, which release cytokines that mimic human inflammatory bowel disease. This suggests that serotonin may cause colorectal tumors. More research is needed to understand how serotonin causes colorectal cancer.

### 9.2. Carcinoid Tumors (CSs)

Carcinoid tumors are among the best described neuroendocrine tumors (NETs), and CS is the most common NET ectopic hormone syndrome. Although many potential mediators have been reported to cause clinical signs of CS, the mediator most commonly used for laboratory confirmation is 5-HIAA, a product of 5-HT degradation. In many cases, increased serotonin secretion has been reported [63,64,67]. Carcinoid tumors account for approximately 1% of all carcinoid malignancies, but their incidence has been increasing in recent years, and carcinoid tumors most commonly occur in the gastrointestinal tract (66%) and bronchopulmonary system (31%) [68]. Targeting the 5-HT receptor may be a complementary receptor and may be a complementary strategy for the treatment of carcinoid tumors, and the use of the 5-HT2B, 2C, receptor antagonist Tegretol may have beneficial effects in the treatment of carcinoid tumors and other diseases [63,64,69].

### 9.3. Cholangiocarcinoma

Increased serotonin synthesis has been observed in cholangiocarcinoma both in vitro and in vivo. Human CC cell lines were also found to express all 5-HT receptor subtypes. The specific inhibition of 5-HT1A, 2A, 2B, 4 and 6 receptors were associated with antiproliferative effects. In addition, the inhibition of serotonin synthesis also inhibits the growth of CC cell lines. This provides a promising target for the future treatment of cholangiocarcinoma [63,64,70].

### 9.4. Hepatocellular Carcinoma (HCC)

There is increasing evidence that serotonin is associated with many pathological conditions in the liver. Serotonin has been found to promote cell survival and the proliferation of the human hepatocellular carcinoma cell lines Huh7 (a liver cancer cell line) and HepG2 (a liver cancer-derived cell line) in a dose-dependent manner [63]. These effects of serotonin are mediated through the 5-HT receptor. In HCC patients, the expression of both 5-HT1B and 5-HT2B receptors is higher than 30%. Both 5-HT1B and 5-HT2B receptors are associated with Ki67, an in vivo marker that indicates the proliferative ability of cancer cells. Its increase also correlates with tumor size. Their antagonists showed potent cytotoxic effects on HepG2 cell lines derived from hepatocellular carcinoma, suggesting that 5-HT receptors may be a novel early therapeutic target for HCC patients [63,64,71].

In recent years, the number of patients with non-alcoholic fatty liver disease (NAFLD) has been increasing worldwide and is closely related to the development of a variety of conditions, including diabetes mellitus, hormonal abnormalities and genetic susceptibility. Its presence in NAFLD, a pathological condition that progresses to cirrhosis and hepatocellular carcinoma, is well established and of interest. The definitive etiology of non-alcoholic fatty liver disease (NAFLD) has not been elucidated, and recent studies have shown that serum serotonin levels decrease in patients with NAFLD as the disease progresses [72]. The hepatic–gut–brain (neural) axis plays a role in NAFLD progression via serotonin and the serotonin receptor HTR2A in hepatocytes, suggesting that HTR2A antagonists are potential therapeutic agents for NAFLD.

Autonomic signaling pathways that connect different organs such as the liver, brain, and gut with 5-HT are important in the pathology of NAFLD and as therapeutic targets, suggesting that serotonin may be an indicator of disease progression. This would also expand the potential of serotonin as a biomarker as well as a diagnostic and therapeutic approach. When cancer is identified early, it is more likely to respond to treatment and can result in a greater probability of survival with less morbidity and less costly treatment. The lives of cancer patients can be significantly improved by detecting cancer early and avoiding delays in care (Figure 3).

## 10. 5-HT Maintenance of Gut Homeostasis

5-HT is a remarkable molecule with a wide range of effects on human bodily functions. Moreover, 95% of 5-HT is produced in the gut, where it plays an important role in the regulation of the enteric nervous system (ENS), immune response and epithelial integrity [73]. It also plays a key role in maintaining gut homeostasis. A growing number of studies have demonstrated the importance of 5-HT in the gut but have also shown us that it is extremely complex. For example, functional gastrointestinal disorders (e.g., irritable bowel syndrome) are now defined as “disorders of gut–brain interactions” [73,74]. This definition suggests a complex etiology. The microbial, epithelial, and immune components of the gut respond to 5-HT signaling but also use 5-HT in signaling pathways that affect gut homeostasis [75,76,77]. A further exploration of the complex 5-HT signaling network in the gut may provide new insights for the benefit of patients.

## 11. Discussion

More than 70 years have passed since the discovery of 5-HT, and several studies have shown that 5-HT plays an important role in gut–brain interactions.

Depression is a common mental disorder, with an estimated 5% of adults worldwide suffering from depression, which can lead to suicide, according to the World Health Organization (WHO, 2023). The mechanisms underlying selective serotonin reuptake inhibitors (SSRIs) are still poorly understood in current research, and there are no reliable methods for predicting treatment response [78].

IBS continues to have a high global prevalence. It is well known that IBS is often complicated by psychiatric and mood disorders [31,32]. A single-arm meta-analysis showed that IBS-C (constipation) had the highest prevalence of depression (38%) and anxiety (40%), followed by IBS-D (diarrhea), IBS-M (mixed symptoms of constipation and diarrhea), and IBS-U (unclassified) [79]. Proper management of IBD is important for disease prognosis. Researchers have studied non-invasive serum biomarkers to identify markers for disease diagnosis, monitoring, and predicting treatment outcomes. Despite extensive research, there are no ideal serum biomarkers for IBD [46,80].

Fortunately, recent studies have also mentioned that 5-HT and serotonin can be used as biomarkers for disease prognosis [33,37,58,72]. Biomarkers are becoming increasingly important in the field of personalized medicine. Serotonin is a biomarker of gut–brain interactions, and its introduction into routine clinical practice is crucial for early diagnosis and treatment.

Biomarkers are defined as “the substances, structures, or processes that can be quantified in the body or its products that influence or predict the occurrence of outcomes or diseases” [81]. Examples of some significant clinical biological markers are blood pressure and pulse rate. The normal reference range for serum serotonin is from 0 to 200 ng/mL, (therapeutic range 9–200 ng/mL) [47,82]. Further extensive experimentation is needed to understand and demonstrate effective normal reference ranges for serum serotonin in individual patients to enable an individualized approach to the early diagnosis, prognosis and therapeutic management of gut–brain disorders.

Serotonin biomarkers can transform the practice of medicine into personalized medicine, helping to diagnose and provide the best treatment for an individual to achieve good outcomes. Of course, good communication between the patient and the doctor is also crucial for the prevention and early treatment of diseases (Figure 4).

The multifactorial pathology and pathogenesis of gut–brain disorders are unknown. It may still not be enough to consider the correlation with serotonin alone, and for the test to be valid, the influence of various factors must also be considered, such as nutrition, diet, physical activity, etc., as well as the relationship with other molecules such as dopamine, histamine, etc. Research using serotonin as a gut–brain biomarker is just beginning to gain a foothold, but it holds great promise for future clinical practice, and the combining of different biomarkers may improve the performance of disease assessment.

The aim of this review is to use 5-HT as a biomarker for the prevention of gut–brain diseases and to contribute to early diagnosis, effective treatment and prognosis management.

In conclusion, the use of serotonin as a biomarker for gut–brain and body-wide organ communication may lead to the development of preventive and innovative treatments for gut–brain disorders, with improved visibility and therapeutic monitoring. It could also be used to gauge stress intensity for self-care and mental health improvement. In the future, the standardization and normalization of testing methods should be harmonized as soon as possible.

## Figures and Tables

**Figure 1 life-14-01280-f001:**
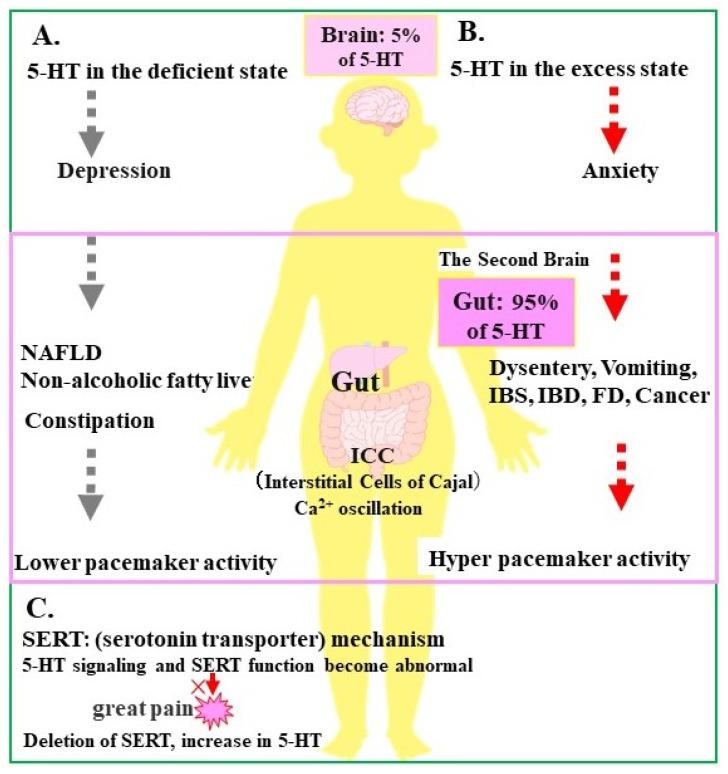
**5-HT receptors and primary functions and SERT mechanism.** (A) 5-HT in the deficient state; (B) 5-HT in the excess state; (C) (serotonin transporter) mechanism for 5-HT signaling and SERT function become abnormal.

**Figure 2 life-14-01280-f002:**
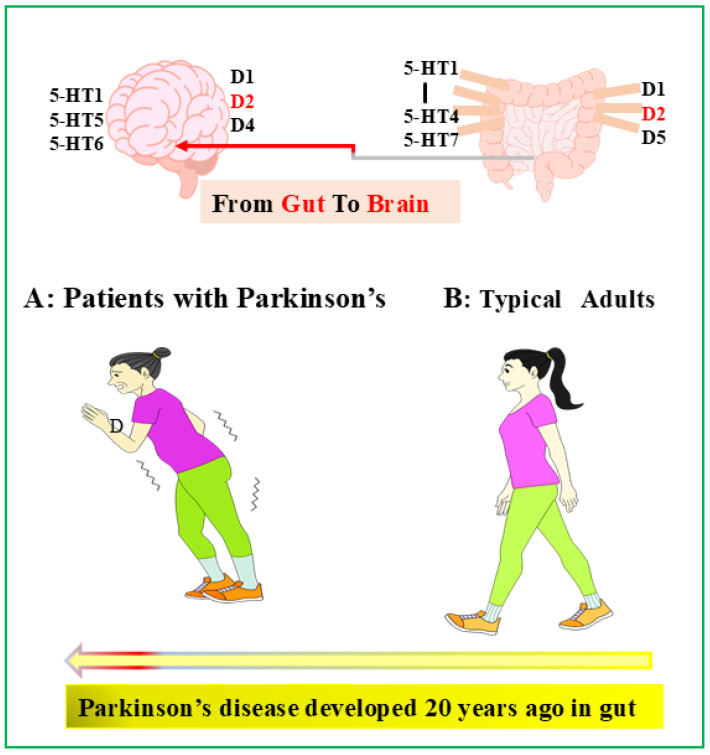
**Parkinson’s disease developed 20 years ago in Gut**. Dopamine receptors (D_1_, D_2_, D_3_, D_4_, and D_5_): D_1_, D_2_, and D_5_ receptors are expressed in the GI tract, and D_1_, D_2_, D_4_ receptors are expressed in the brain.

**Figure 3 life-14-01280-f003:**
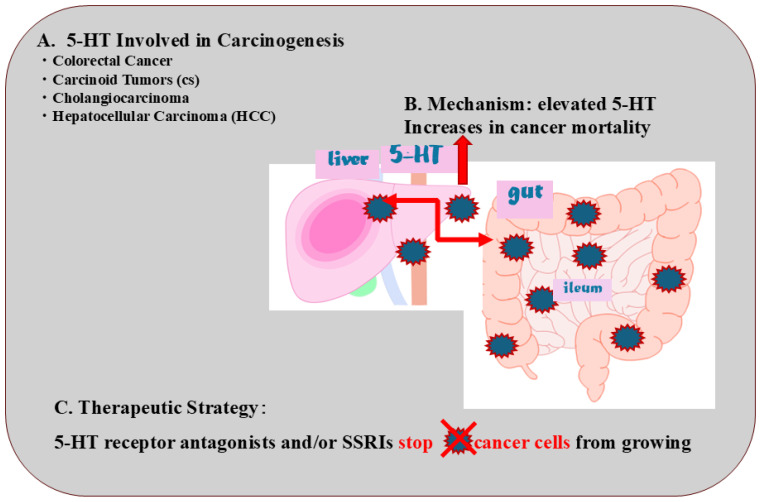
**Role of serotonin in carcinogenesis.** (A) 5-HT involved in carcinogenesis; (B) mechanism: elevated 5-HT increases cancer mortality risk; (C) therapeutic strategy: 5-HT receptor antagonists and/or SSRIs stop cancer cells from growing.

**Figure 4 life-14-01280-f004:**
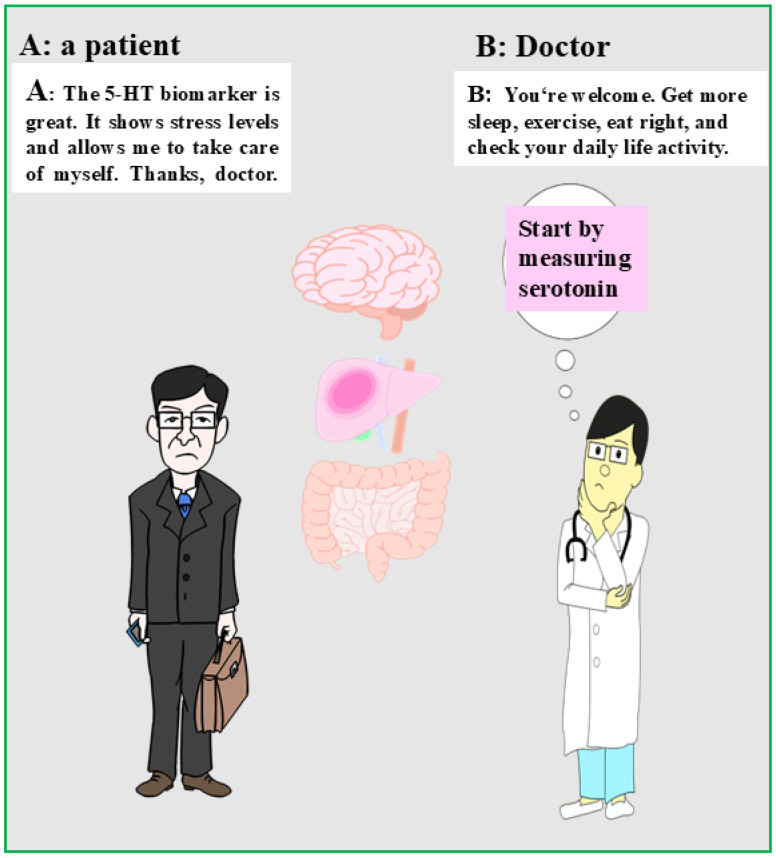
Good communication between patient and doctor.

**Table 1 life-14-01280-t001:** Latest trends in biomarkers for gut–brain interactions.

Role of Serotonin (5-Hydroxytryptamine: 5-HT), a Neurotransmitter That Regulates Mood in the Brain and Signaling in the Gut			
Biomarkers	Clinical Data Findings	Preclinical Data Findings	References
1. A: Depression B: IBS, IBD	A: 12 B: 26		A: [1,2,3,4,5,39,40,41,42,43,44,45] B: [3,14,15,16,17,18,19,20,21,22,23,24,25,31,32,33,34,35,36,37,38,46]
2. Serotonin Toxicity	3		[6,39,47]
3. 5-HT Receptors and Primary Functions	8	3	[3,6,7,8,9,10,11,12,28,29,30]
4. Co-operation between Serotonin and Dopamine	7	3	[48,49,50,51,52,53,54,55,56,57]
5. Endoscopy and 5-HT as a Gut–Brain Biomarker	5		[4,5,33,46,58]
6. Functional Dyspepsia (FD) and 5-HT receptor	4		[59,60,61,62]
7. 5-HT Involved in Carcinogenesis	8	2	[63,64,65,66,67,68,69,70,71,72]

**Table 2 life-14-01280-t002:** Gut–brain 5-HT functional distribution.

5-HT Receptors	Subtype	Location/Function
		PNS: myenteric plexus, smooth muscle, m-RNA in vascular tissue
**5-HT1**	5-HT1A, 5HT1B, 5-HT1D, 5-HT1E, 5-HT1F	CNS: hippocampus, subiculum substantial nigra, cranial blood vessel
		PNS: GI, vascular, smooth muscle, platelets
**5HT2**	5-HT2A, 5-HT2B, 5-HT2C	CNS: cerebral cortex, cerebellum hypothalamus, hippocampus
		PNS: Abdominal visceral afferent neuron, ICC cell, gut
**5HT3**	5-HT3A, 5-HT3B, 5-HT3C, 5-HT3D, 5-HT3E	CNS: area postrema
		PNS: GIT, smooth muscle
**5HT4**	5-HT4A, 5-HT4B, 5-HT4C, 5-HT4D	CNS: hippocampus
**5HT5**	5-HT5A, 5-HT5B	CNS: olfactory bulb, Habenula
		PNS: Superior cervical ganglia
**5HT6**		CNS: caudate putamen, hippocampus
		PNS: gastrointestinal, vascular smooth muscle
**5HT7**		CNS: hypothalamus

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
