# Peer review of "New Role of the Serotonin as a Biomarker of Gut–Brain Interaction"

_life, 2024, doi:10.3390/life14101280_

Round 1
Reviewer 1 Report
Comments and Suggestions for Authors
Dear Authors,
Your article entitled " New Role of the Serotonin as a Biomarker of Gut-Brain Interaction" has been reviewed.
Kindly find below the list of my comments:
The review supposes that serotonin has potential to be a valuable biomarker for the relationships between the gut and brain in both IBS and IBD. Overall, the review makes a compelling case for further research into the use of serotonin as a biomarker for gastrointestinal disease and may even find application in mental health testing by measuring stress intensity.
But to fulfill these intentions, the text must offer a more comprehensive and clinically relevant discussion of serotonin syndrome and serotonin toxicity.
Although the text mentions the different types of serotonin receptors, it does not explore the nuances of how these receptors may differentially influence the pathophysiology of various diseases.
Additionally, there is a lack of discussion of the challenges associated with using serotonin as a biomarker, including its specificity and sensitivity compared to other existing biomarkers.
Best Regards,
Reviewer 2 Report
Comments and Suggestions for Authors
Dear All,
Thank you for the opportunity to review this paper. This paper presents a novel and compelling argument for the use of serotonin as a biomarker for gut-brain interactions, highlighting its potential applications in diagnosing and managing a range of gastrointestinal and psychological disorders. The authors have successfully synthesized a wide array of literature, bringing together diverse studies to build a comprehensive overview of serotonin's multifaceted roles within both the central nervous system and the gastrointestinal tract. The manuscript is well-organized, and the authors have clearly articulated the current gaps in the literature, as well as the potential benefits of using serotonin as a non-invasive biomarker for gut-brain communication. For these reasons, I think that this paper could be taken into consideration for publication in Life.
Reviewer 3 Report
Comments and Suggestions for Authors
This review manuscript has scientific merit that might benefit readers, but some major revisions still need to be included. Therefore, we recommend that the authors make the following modifications:
- Line 9-10: Abstract: "Serotonin (5-hydroxytryptamine: 5-HT), a neurotransmitter that regulates mood in the brain and signaling in the gut, has receptors throughout the body that serve various functions."
- The authors are suggested to add at least two specific functions that these receptors have, as it would make this statement more informative.
- Line 24-25: "5-HT, a monoamine neurotransmitter involved in mood control in the brain, also plays a critical role in cellular signaling in the gut."
- The authors are suggested to explain or provide examples of the type of cellular signaling in which serotonin is involved, which would make this statement more comprehensive.
- The authors are suggested to create a table in the Methods section to demonstrate:
Table 1. Latest trends in biomarkers for gut-brain interactions
Role of Serotonin (5-hydroxytryptamine: 5-HT), a neurotransmitter that regulates mood in the brain and signaling in the gut |
|||
Biomarkers |
Clinical data findings |
Preclinical data findings |
Ref. |
1. Depression |
|
|
|
2. Serotonin Toxicity |
|
|
|
3. 5-HT Receptors and Primary Functions |
|
|
|
4. Co-operation between serotonin and dopamine |
|
|
|
5. Endoscopy and 5-HT as a gut-brain biomarker |
|
|
|
6. Functional Dyspepsia (FD) and 5-HT receptor |
|
|
|
7. 5-HT Involved in Carcinogenesis |
|
|
|
- Line 349:
Section 9. 5-HT Involved in Carcinogenesis
The authors are suggested to include a schematic diagram as Figure 3. Role of Serotonin in Carcinogenesis, illustrating the connection between serotonin and the following cancers:
- Colorectal Cancer
- Carcinoid Tumors (CS)
- Cholangiocarcinoma
- Hepatocellular Carcinoma (HCC)
This figure should depict the mechanism of serotonin’s growth-stimulatory effect on these cancers and carcinoids. Including this would strengthen the manuscript, as the link between 5-HT and cancer mechanisms adds significant interest, especially considering the severe mortality rates associated with cancer.
Round 2
Reviewer 1 Report
Comments and Suggestions for Authors
Dear Authors,
I think the corrections made complement your idea of ​​using serotonin as a biomarker.
Regards!
Reviewer 3 Report
Comments and Suggestions for Authors
The authors have addressed all the comments with logical explanations. We wish the authors the best of luck.
Best wishes.